# Ginsenoside Rb_1_ Ameliorates Heart Failure Ventricular Remodeling by Regulating the Twist1/PGC-1α/PPARα Signaling Pathway

**DOI:** 10.3390/ph18040500

**Published:** 2025-03-30

**Authors:** Ziwei Zhou, Zhimin Song, Xiaomeng Guo, Qi Wang, Meijing Li, Minyu Zhang, Muxin Gong

**Affiliations:** 1School of Traditional Chinese Medicine, Capital Medical University, Beijing 100069, China; zhouziwei199509@163.com (Z.Z.);; 2Beijing Key Laboratory of Traditional Chinese Medicine Collateral Disease Theory Research, Beijing 100069, China

**Keywords:** heart failure, Twist homolog 1 (Twist1), ginsenoside Rb_1_, myocardial energy metabolism disorder

## Abstract

**Background:** Heart failure (HF), the terminal stage of cardiovascular disease with high morbidity and mortality, remains poorly managed by current therapies. Ventricular remodeling in HF is fundamentally characterized by myocardial fibrosis. While ginsenoside Rb_1_ has demonstrated anti-fibrotic effects in HF, the underlying mechanism remains unclear. Twist1, an upstream regulator of energy metabolism factors PGC-1α and PPARα, may attenuate fibrosis by preserving systemic energy homeostasis, suggesting its pivotal role in HF pathogenesis. This study explores ginsenoside Rb_1_′s anti-HF mechanisms through the regulation of ginsenoside Rb_1_ on these metabolic regulators. **Methods:** Sprague Dawley rats were subjected to a ligation of the left anterior descending coronary artery to induce an HF model, followed by ginsenoside Rb_1_ treatment for 6 weeks. Therapeutic effects were evaluated through cardiac function assessment, myocardial histopathological staining (HE, Masson, immunofluorescence, immunohistochemistry), mitochondrial morphology observation (transmission electron microscopy), energy metabolism analysis (electron transport chain efficiency, mitochondrial membrane potential, ATP content), and protein expression profiling (Twist1, PGC-1α, PPARα, GLUT4, PPARγ). Additionally, H9c2 cells induced with endothelin-1 to model HF were employed as an in vitro model to further investigate ginsenoside Rb_1_′s regulatory effects on the Twist1/PGC-1α/PPARα signaling pathway. **Results:** Ginsenoside Rb_1_ can restore cardiac function in HF rats, improve mitochondrial function, alleviate energy metabolism disorders, and inhibit ventricular remodeling. By modulating the Twist1/PGC-1α/PPARα signaling pathway, ginsenoside Rb_1_ suppressed the abnormal overexpression of Twist1 and maintained normal expression of downstream PGC-1α and PPARα. In vitro experiments further demonstrated that ginsenoside Rb_1_ significantly inhibited Twist1 expression in H9c2 cardiomyocytes with HF while promoting PGC-1α and PPARα expression, thereby restoring myocardial energy metabolism and mitigating ventricular remodeling in HF. **Conclusions:** Ginsenoside Rb_1_ can inhibit the upregulation of Twist1 and activate the expression of its downstream PGC-1α and PPARα expression, by modulating the Twist1/PGC-1α/PPARα signaling pathway, alleviating ventricular remodeling in HF patients and improving myocardial energy metabolism dysfunction. Twist1 may be a key target for the treatment of HF. This study not only elucidates the mechanism by which ginsenoside Rb_1_ alleviates HF, but also provides new insights into the clinical treatment of HF.

## 1. Introduction

Heart failure (HF) is a complex clinical syndrome with symptoms and/or signs caused by insufficient cardiac output resulting from any structural and/or functional abnormalities of the heart [1,2]. HF is supported by elevated natriuretic peptide levels and/or objective evidence of pulmonary or systemic congestion, as well as high morbidity and mortality and poor quality of life. It also incurs substantial costs [1]. In 2017, more than 64 million individuals around the globe were diagnosed with HF [3]. The estimated actual prevalence of HF is approximately 2% in Europe and 2.4% in the United States, with expectations for this rate to increase to 3% by 2030 [4]. Current estimates suggest that there are 10 million HF patients in our country, with the prevalence still rising [5]. Mitochondrial energy damage and energy metabolism disorder are important characteristics of HF, which lead to insufficient myocardial productivity and induce ventricular remodeling in HF [6]. Despite groundbreaking advances in guideline-directed medical therapy and remarkable improvements in therapeutic outcomes, the epidemiological burden of HF continues to escalate, with persistently rising incidence and mortality rates globally. Current therapies targeting HF-related energy metabolism disorders face challenges, such as unclear mechanisms of action and oversimplified intervention strategies. Their limited efficacy leaves most patients without optimal treatment [7,8,9]. Chinese medicine represented by *Panax ginseng* has the function of regulating cardiac energy metabolism, which may be a safer and more effective treatment for HF.

*Panax ginseng* was first documented in “*Shen Nong’s Herbal Classic*”, is known as the “King of Herbs”, and is a representative traditional Chinese medicine for tonifying heart qi and treating HF [10]. Ginsenoside Rb_1_, a major active component of *Panax ginseng* (Figure 1), exhibits broad pharmacological effects. In previous studies in rat models of myocardial infarction (MI)-induced HF, ginsenoside Rb_1_ restored cardiac function and alleviated HF through multi-pathway and multi-target mechanisms, including mitigating oxidative stress, exerting anti-inflammatory effects, and improving mitochondrial quality control. Du Lixin et al. demonstrated that ginsenoside Rb_1_- PLGA nanoparticles significantly improved myocardial oxidative stress damage and pathological conditions in HF rats by activating the reactive oxygen species (ROS)/peroxisome proliferator-activated receptor alpha (PPARα)/peroxisome proliferator-activated receptor gamma coactivator 1α (PGC-1α) pathway [11]. Meanwhile, Zheng Zhi’s team developed a biomimetic adhesive-injectable hydrogel targeting mitochondrial DNA (mtDNA)-STING signaling crosstalk, which enhanced inflammation clearance and mitochondrial repair to promote MI recovery [12]. Notably, ginsenoside Rb_1_ further improved mitochondrial quality control by regulating the dual-specificity phosphatase-1 (DUSP-1)- BAX inhibitor motif containing 6 (TMBIM6) axis, suppressed inflammatory responses, and modulated gut microbiota to maintain mitochondrial homeostasis, ultimately reversing HF progression [13].

Additionally, ginsenoside Rb_1_ effectively improved cardiac dysfunction in HF mice and isoproterenol-induced HF rats, reducing myocardial hypertrophy, fibrosis, and collagen deposition. It restored energy metabolism by regulating cardiac enzyme activity and fatty acid oxidation, promoting ATP production, maintaining mitochondrial function, and delaying HF progression [14,15]. However, the molecular mechanisms by which ginsenoside Rb_1_ inhibits cardiac fibrosis in MI-induced HF rats via improving energy metabolism remain unclear.

Twist homolog 1 (Twist1) belongs to the family of basic helix–loop–helix transcription factors and plays a pivotal role in heart valve development by promoting cell proliferation, migration, and the expression of extracellular matrix-related genes [16]. The relationship between energy metabolism disorders and ventricular remodeling in HF has been established in early research. PGC-1α and PPARα are recognized as key factors to control energy homeostasis, which affect myocardial energy metabolism by regulating mitochondrial biosynthesis, fatty acid oxidation, and glucose metabolism [17,18]. The latest research shows that Twist1 is an upstream regulator of PGC-1α and PPARα, which plays a significant role in regulating energy metabolism. According to reports, Twist1 increases under chronic hypoxia and pathogenic conditions. Blocking the expression of Twist1 can alleviate mitochondrial dysfunction and intracellular lipid accumulation and promote the expression of PGC-1α and downstream target genes. It is worth noting that Twist1 has the function of regulating energy metabolism and inhibiting fibrosis, and can be used as a potential intervention target for anti-fibrosis treatment [19]. However, how Twist1 plays a role in HF ventricular remodeling characterized by myocardial fibrosis has not been clarified.

Therefore, we regard Twist1 as the starting point, combined with its regulatory effects on PGC-1α and PPARα, to verify the regulatory role of ginsenoside Rb_1_ in heart fibrosis caused by energy metabolism imbalance in HF through interfering with Twist1, with a view to providing potential sources for new diagnostic and prognostic biomarkers of energy metabolism disorder in HF.

## 2. Results

### 2.1. Ginsenoside Rb_1_ Can Improve Cardiac Function Damage in Rats with HF

Echocardiography was employed to assess whether ginsenoside Rb_1_ altered cardiac function in rats with HF following MI. The results indicated that in the HF group, the ejection fraction (EF) and fractional shortening (FS) were reduced (*p* < 0.05 or *p* < 0.01), while the LV end-diastolic and end-systolic internal diameters (LVID(d) and LVID(s)), as well as the LV end-diastolic and end-systolic volumes (LV vol (d) and LV vol (s)), were increased (*p* < 0.05 or *p* < 0.01). Additionally, the LV end-diastolic and end-systolic anterior wall thicknesses (LVAW(d) and LVAW(s)), as well as the LV end-diastolic and end-systolic posterior wall thicknesses (LVPW(d) and LVPW(s)), were significantly decreased (*p* < 0.05 or *p* < 0.01) (Figure 2C,D); the heart weight index (HWI) was significantly increased (*p* < 0.05 or *p* < 0.01 or *p* < 0.001) (Figure 2A), indicating left ventricular enlargement, abnormal cardiac structure, and weakened cardiac contractile function. However, after the administration of ginsenoside Rb_1_ or valsartan, EF and FS increased, LVAW(d) and LVAW(s) increased, LVPW(d) and LVPW(s) increased, LVID(d) and LVID(s) decreased, and LV Vol(d) and LV Vol(s) decreased. Natriuretic peptides are widely used in the diagnosis and evaluation of HF and are representative biomarkers of myocardial injury, closely related to the prognosis of HF [20]. Therefore, we detected the expression of atrial natriuretic peptide (ANP) and brain natriuretic peptide (BNP). Biochemical analysis showed that compared to the HF group, ginsenoside Rb_1_ administration significantly inhibited the abnormal increase in ANP and BNP levels (*p* < 0.05 or *p* < 0.01 or *p* < 0.001) (Figure 2B). These results suggest that ginsenoside Rb_1_ can improve cardiac contractile function and reduce myocardial injury in HF rats.

### 2.2. Ginsenoside Rb_1_ Can Improve the Pathological Changes in the Left Ventricle of Rats with HF

To further confirm the role of ginsenoside Rb_1_ in in vivo models, we observed the morphological changes in rat heart tissues. The results showed that ginsenoside Rb_1_ significantly attenuated the enlargement of the heart volume in rats with HF (Figure 3A). HWI was also reduced in rats with HF. The pathological staining results indicated that compared to the sham group, rats with HF exhibited a significant increase in inflammatory cell infiltration and pale collagen deposition in the left ventricle (Figure 3B), with myocardial fibrosis and the excessive accumulation of blue collagen fibers in the left ventricular wall (Figure 3C), as well as an enlarged cross-sectional area. However, these pathological changes were significantly reversed after the administration of ginsenoside Rb_1_, especially in the HF+Rb_1_-H group. These findings suggest that ginsenoside Rb_1_ can effectively alleviate the morphological pathological changes in rats with HF.

### 2.3. Ginsenoside Rb_1_ Can Improve the Myocardial Mitochondrial Ultrastructure in Rats with HF

To clarify the ultrastructural characteristics of impaired myocardial mitochondrial bioenergetics, we conducted an assessment using transmission electron microscopy. The results revealed that in the sham group, the myocardial mitochondria exhibited normal morphology and size, with a clear and compact structure, and the Z-lines were neatly arranged. Compared to the sham group, the HF group showed myocardial mitochondrial damage with irregular morphology, mitochondrial swelling, and the fragmentation and loose dissolution of mitochondrial cristae, and some were accompanied by a reduction in matrix particles or even vacuolization. After the administration of valsartan and ginsenoside Rb_1_, the myofibrils were densely and regularly arranged, with only a small number of myofibril ruptures. Especially in the HF+Rb_1_-H group, the Z-lines were neatly arranged and the morphology was essentially normal (Figure 3D).

### 2.4. Ginsenoside Rb_1_ Can Reduce the Expression of cTnT and ACTN2 in Rats with HF

Cardiac troponin T (cTnT) is a contractile protein in cardiomyocytes that is widely used to assess chronic heart disease; it can reflect myocardial cell damage and is a biomarker for the risk stratification of myocardial failure [21,22]. Actin α2 (ACTN2) is an actin-binding protein that has multiple roles in different cell types. Located on the Z disk of the sarcomere, it plays a connecting role between antiparallel actin filaments and regulates the systolic and diastolic functions of the heart [23,24]. After MI, the structure and function of the heart undergo rapid changes. We detected the expression of representative biomarkers cTnT and ACTN2 to further elucidate the improvement effect of ginsenoside Rb_1_ on cardiac injury and fibrosis. Immunofluorescence results showed that the positive expression of cTnT and ACTN2 proteins was enhanced in the HF group, which was reversed after the administration of ginsenoside Rb_1_ (*p* < 0.01 or *p* < 0.001) (Figure 4A). Additionally, the ELISA method was used to detect the expression level of cTnT in the serum of rats (Figure 4B), and the results were consistent with those obtained by immunofluorescence. Ginsenoside Rb_1_ can inhibit the elevation of cTnT in the serum. The above results indicate that ginsenoside Rb_1_ has an inhibitory effect on cardiac injury caused by HF after MI, protecting the heart’s systolic and diastolic functions.

### 2.5. Ginsenoside Rb_1_ Can Improve Myocardial ATP Content, ETC Productivity Efficiency, and MMP Level in Rats with HF

To verify the integrity of mitochondrial homeostasis and function after mitochondrial damage, the effect of ginsenoside Rb_1_ on cardiac energy metabolism in rats with HF was evaluated, and ATP content, ETC productivity efficiency, and MMP level were measured. The results show that ginsenoside Rb_1_ could increase myocardial ATP content (*p* < 0.05 or *p* < 0.01), ETC productivity efficiency (*p* < 0.001), and MMP level in rats with HF (*p* < 0.01 or *p* < 0.001), ensuring cardiac energy supply, maintaining mitochondrial homeostasis and functional integrity, and alleviating HF (Figure 4C–E).

### 2.6. Ginsenoside Rb_1_ Improves Myocardial Energy Metabolism in HF by Inhibiting Twist1 and Activating PGC-1α

To investigate the effects of ginsenoside Rb_1_ on myocardial energy metabolism in rats with HF by regulating the Twist1/PGC-1α/PPARα signaling pathway, WB was performed to verify the expression of Twist1, PGC-1α, and PPARα in rat myocardium. The results showed that ginsenoside Rb_1_ could significantly reverse the upregulation of Twist1 (*p* < 0.05 or *p* < 0.001) and the downregulation of PGC-1α and PPARα (*p* < 0.05 or *p* < 0.001) in rat myocardium. Additionally, GLUT4 and PPARγ were significantly decreased in the myocardium of rats with HF, while ginsenoside Rb_1_ inhibited the decrease in GLUT4 and PPARγ (*p* < 0.01 or *p* < 0.001) (Figure 5A,B). Subsequently, immunohistochemical staining methods were used to detect the expression of Twist1, PGC-1α, and PPARα in the myocardium of rats (Figure 5C–E). In the HF model group, the abnormal upregulation of Twist1 and the downregulation of PGC-1α and PPARα were significantly inhibited by ginsenoside Rb_1_.

### 2.7. Ginsenoside Rb_1_ Can Improve the ATP Content, ETC Productivity Efficiency, and MMP Level in H9c2 Cells

The results of in vitro experiments demonstrate that ginsenoside Rb_1_ can increase ATP content (*p* < 0.05, *p* < 0.01 or *p* < 0.001), ETC productivity efficiency (*p* < 0.01 or *p* < 0.001), and MMP level (*p* < 0.05, *p* < 0.01 or *p* < 0.001) in injured H9c2 cells, ensuring cardiomyocyte energy supply (Figure 6A,B).

### 2.8. Ginsenoside Rb_1_ Improves the Energy Metabolism of Injured H9c2 Cells by Regulating the Twist1/PGC-1α/PPARα Signaling Pathway

To further elucidate the mechanisms of action in both low-dose and high-dose ginsenoside Rb_1_ groups, we conducted additional validations at the cellular level. The results found that ginsenoside Rb_1_ significantly reversed the endothelin-1 (ET-1)-induced upregulation of Twist1 (*p* < 0.05, *p* < 0.01 or *p* < 0.001), inhibited the downregulation of PGC-1α and PPARα (*p* < 0.05, *p* < 0.01 or *p* < 0.001), and alleviated cardiomyocyte injury and energy metabolism disorders (Figure 7).

### 2.9. Molecular Docking

The molecular docking results between ginsenoside Rb_1_ and Twist1 were shown in Figure 8A. The affinity energy between ginsenoside Rb_1_ and Twist1 were calculated as −6.343 kcal/mol, indicating significant intermolecular affinity and high stability in the binding conformation. Furthermore, ginsenoside Rb_1_ can bind to the amino acid residues PRO139, THR137, LYS133, LEU143, ARG132, and LYS145 of Twist1 through hydrogen bonding or hydrophobic interactions, thereby affecting the conformational stability or DNA-binding ability of Twist1 (Figure 8B). These findings suggest that ginsenoside Rb_1_ exerts anti-HF effects by specifically targeting and modulating Twist1.

## 3. Discussion

After MI, the myocardium experiences ischemia and hypoxia, leading to a decline in heart function, which subsequently results in HF [25]. Ventricular remodeling is the core pathological alteration in HF, with myocardial fibrosis serving as its fundamental pathological basis. A significant pathological change contributing to ventricular remodeling is cardiac metabolic remodeling [26], which involves alterations in myocardial substrate metabolism and decreased energy production. Mitochondria serve as the primary site for myocardial energy production and metabolism, accounting for 30% of the total volume of cardiomyocytes [7]. Mitochondrial dysfunction and disordered energy metabolism lead to “insufficient energy supply, homeostasis disruption, and metabolic disturbances” in the heart, triggering abnormal structural and functional changes in the heart and ultimately contributing to the development of HF [27,28]. Therefore, targeting mitochondrial bioenergetics and altering energy metabolism in the hearts of HF patients may represent an attractive strategy to alleviate symptoms and improve cardiac function in these individuals.

The classic Chinese medical text “*Shen Nong’s Herbal Classic*” records that ginseng has the effects of tonifying qi and strengthening the pulse [29], while HF is a clinical manifestation of “qi deficiency of the heart” in traditional Chinese medicine [30]. Therefore, ginseng is often used to treat HF. Ginsenoside Rb_1_, the primary active ingredient in *Panax ginseng*, is a natural extract with multiple cardioprotective effects. It plays a protective role in various cardiovascular diseases through mechanisms such as improving energy metabolism, inhibiting autophagy, and exerting anti-inflammatory effects [31]. Previous studies have shown that ginsenoside Rb_1_ can reduce lipid deposition and mitochondrial damage in diabetic mice, as well as improve energy homeostasis and cardiac function [32]. It enhances MMP, maintains mitochondrial homeostasis, and stimulates mitochondrial biosynthesis [13]. Additionally, ginsenoside Rb_1_ alleviates mitochondrial dysfunction by improving glucose uptake [33]. However, it is unclear whether ginsenoside Rb_1_ can alleviate ventricular remodeling in HF by improving these factors. Consequently, we investigated the role of ginsenoside Rb_1_ in modulating myocardial energy metabolism to improve ventricular remodeling in HF.

PPARs are lipid sensors that regulate energy metabolism throughout the body, including the heart. PPARα can enhance cellular fatty acid uptake, esterification, and transport, while regulating lipoprotein metabolism genes [34]. As a transcriptional coactivator of PPARs, PGC-1α interacts with PPARα and PPARγ to regulate the expression of mitochondrial genes, indirectly promoting the transportation and utilization of fatty acids, and playing a key role in controlling myocardial metabolism [35]. An increasing number of studies suggest that Twist1 is closely related to lipid metabolism in fat metabolism [36]. Experimental studies have found that the overexpression of Twist1 inhibits the expression of PGC-1α target genes and mitochondrial biogenesis, functioning as a metabolic brake for energy expenditure in brown adipose tissue and being vital for maintaining whole-body energy homeostasis [37]. It is also involved in regulating fatty acid metabolism, thereby modulating the transition from white adipose tissue to brown adipose tissue [38]. In another study, it was also proved that Twist1 overexpression activated the pentose phosphate pathway, glycolysis, etc., which was closely related to glucose metabolism [39]. Therefore, Twist1 has become an attractive target for the treatment of HF caused by energy metabolism.

On the other hand, Twist1 plays an important role in various fibrotic diseases; it is highly expressed in the heart, renal, skin, and oral mucosa [36,40,41,42]. Research has found that in mouse models, continuous expression in heart valves promotes cell proliferation and disrupts fibrous collagen, and a high expression of Twist1 has been observed in human diseased aortic valves [43]. In addition, activating Twist1 promotes the proliferation and migration of cardiac fibroblasts, accelerating the progression of cardiac fibrosis [44]. Evidently, Twist1 mediates the occurrence of heart disease and the progression of cardiac fibrosis, plays an important role in heart disease, and is a promising target for the treatment of HF. In summary, Twist1, as an emerging therapeutic target influencing ventricular remodeling in HF, not only serves as a potential intervention site for fibrosis but also acts as a critical regulator of energy metabolism. Therefore, we focused on the Twist1/PGC-1α/PPARα signaling pathway to investigate the ameliorative effects of ginsenoside Rb_1_ on cardiac energy metabolism dysfunction in HF.

We observed that Twist1 expression was upregulated in the hearts of rats with HF and was similarly elevated in the ET-1-induced injury model of H9c2 cells, suggesting that Twist1 mediates the occurrence of HF to a certain extent. Ginsenoside Rb_1_ suppressed the abnormally elevated expression of Twist1 in the hearts of rats with HF, maintained normal expression levels of PGC-1α and PPARα, and alleviated ventricular remodeling. This study suggests that ginsenoside Rb_1_ ameliorates myocardial energy metabolism and ventricular remodeling in HF by modulating the Twist1/PGC-1α/PPARα signaling pathway, specifically through inhibiting Twist1 expression while promoting the expression and activation of PGC-1α and its downstream target PPARα (Figure 9). Notably, our study revealed that Twist1 is highly expressed during ventricular remodeling in HF. This upregulation correlates with myocardial fibrosis. Therefore, monitoring Twist1 levels can help assess fibrosis severity and guide anti-fibrotic therapy. In addition, combining Twist1 with its downstream targets (PGC-1α and PPARα) improves diagnostic accuracy. As a sentinel molecule coordinating early remodeling events, Twist1 shows unique clinical value in monitoring therapeutic responses and guiding targeted interventions. Twist1, as a predictive biomarker and a novel therapeutic target, has broad application prospects in optimizing the treatment of HF.

Future studies should further validate the molecular mechanisms of Twist1 in MI-induced HF through Twist1 overexpression or knockout experiments. We will focus on investigating the regulatory effects of ginsenoside Rb_1_ on Twist1-mediated ventricular remodeling in HF, with a specific emphasis on elucidating its underlying mechanisms. The experimental design includes in vivo administration of ginsenoside Rb_1_ to cardiac-specific Twist1-overexpressing HF rat models, and in vitro treatment of Twist1-overexpressing cardiomyocytes with ginsenoside Rb_1_ to verify its regulatory role in HF pathogenesis. Based on these findings, ginsenoside Rb_1_ could serve as a lead compound for developing targeted inhibitors to suppress Twist1 hyperactivation and biosynthesis, thereby providing novel therapeutic strategies for ventricular remodeling HF.

## 4. Materials and Methods

### 4.1. Drugs

Ginsenoside Rb_1_ (Rb_1_; chemical structure: C_54_H_92_O_23_; molecular weight: 1109.29; CAS 41753-43-9; Cat. Number: PS011946) at a purity of 99.32% was purchased from Chengdu Push Bio-Technology Co., Ltd. (Chengdu, Sichuan, China). The positive drug valsartan was purchased from Tianda Pharmaceutical (Zhuhai) Co., Ltd. (Zhuhai, Guangdong, China).

### 4.2. Animal Models and Experimental Groups

The animal experiments were approved by the Ethics Committee of Capital Medical University (Ethics Approval Number: AEEI-2022-240; Approval Date: 11 April 2023). Male specific pathogen-free (SPF) SD rats (180–200 g) were purchased from Beijing Vital River Laboratory Animal Technology Co., Ltd. [Animal License Number: SCXK (Beijing, China) 2021-0011]. All experimental procedures were conducted in accordance with the “Guidelines for the Care and Use of Laboratory Animals” published by the National Institute of Health.

After one week of adaptive feeding, the rats were randomly divided into a sham group (*n* = 8) and an HF model group. HF after MI was induced surgically in rats by left anterior descending coronary artery, while the sham was only threaded without ligation. Based on the dosage and administration timing of ginsenoside Rb_1_ reported in previous literature for SD rats [15], and combined with our research group’s preliminary studies on the post-surgical model establishment and treatment time in rat with HF following MI [45], we determined the drug dosage and time points used in the current experiment. Four weeks post-surgery, the 32 rats with confirmed HF were randomly divided into 4 groups, and received daily oral gavage treatments for 6 weeks: (1) HF group (equal volume of distilled water; *n* = 8); (2) HF + valsartan group (8 mg/kg [45,46] dissolved in 0.5% sodium carboxymethyl cellulose (CMC-Na); *n* = 8); (3) HF + Rb_1_-L group (40 mg/kg dissolved in 0.5% CMC-Na; *n* = 8); and (4) HF + Rb_1_-H group (80 mg/kg dissolved in 0.5% CMC-Na; *n* = 8). All SD rats were housed in SPF conditions (constant temperature of 20–25 °C, humidity at 50–70%, and a 12 h light/12 h dark cycle) with free access to food and water. Body weight was measured every 7 days. The Animal Ethics Committee of Capital Medical University approved the experimental scheme (No.:110011231103535785). The experiment process conforms to the detailed rules for the implementation of the China State Administration of Medical Laboratory Animals.

### 4.3. Echocardiographic Detection

After taking the drug for 6 weeks, the left ventricular structure and function were evaluated using a Visual Sonics Vevo 2100 high-resolution ultrasound system equipped with a 15 MHz transducer (VisualSonics, Toronto, ON, Canada). EF, FS, LVID(d), LVID(s), LV vol(d), LV vol(s), LVAW(d), LVAW(s), LVPW(d), and LVPW(s) were measured via long-axis two-dimensional, M-mode parasternal echocardiography.

### 4.4. Animal Tissue Harvesting

After weighing and anesthetizing the rats, blood samples were collected via abdominal aorta puncture into a 5 mL vacuum tube without additives. The blood samples were allowed to stand at room temperature for approximately 2 h, and were then centrifuged at 3000 rpm for 10 min to separate the serum.

After blood collection, these rats were executed and their hearts were immediately excised, photographed, and weighed. The heart tissues were then divided into three parts—one part was cut into 1 mm^3^ cubes and quickly placed in glutaraldehyde for transmission electron microscope sample preparation; another part was rapidly frozen in liquid nitrogen for tissue protein and mitochondrial extraction; and the remaining part was placed in 4% paraformaldehyde for the preparation of pathological tissue sections.

### 4.5. Histopathological Examination

Heart tissues from each group were immersed in 4% paraformaldehyde for 24 h and dehydrated by alcohol gradient concentration. The samples were immersed in xylene for 1 h and were subsequently embedded in paraffin. After staining with hematoxylin–eosin (HE) and Masson’s trichrome, the pathological changes in the heart tissues were observed under a microscope (Olympus, Tokyo, Japan).

### 4.6. Immunofluorescence and Immunohistochemistry Staining

Paraffin sections of the left ventricle of the heart were incubated with anti-ACTN2 antibody and anti-cTnT antibody (ab137346, ab8295, 1:1000, Abcam, UK) at 4 °C overnight. After washing, the sections were incubated with fluorescent secondary antibodies (ab150080, ab150113, 1:500, Abcam, Cambridge, UK) for 1 h at room temperature, and were then washed again and incubated with DAPI for 10 min for nuclear staining. Finally, the stained sections were observed using a fluorescence microscope (Wetzlar, Germany).

Paraffin sections of the left ventricle of the heart were incubated with anti-Twist1 antibody (A25134, 1:2000, ABclonal, Wuhan, China), anti-PGC1α antibody (66369-1-1g, 1:500, Proteintech, Wuhan, China), and anti-PPARα antibody (66826-1-1g, 1:500, Proteintech, China) overnight at 4 °C for immunohistochemical. After washing with PBS, the sections were incubated with secondary antibody (G1302, 1:200, Servicebio, Wuhan, China) at room temperature for 1 h following another wash with PBS; DAB (3,3-dia-minobenzidine) was used for color development, followed by hematoxylin staining. After mounting, the sections were observed and photographed using a microscope (Nikon, Tokyo, Japan). Brown particles were identified as positive cells.

### 4.7. Cell Culture

Rat H9c2 cardiomyocytes (Cat# CL-0089, Wuhan, China) were cultured in Dulbecco’s Modified Eagle Medium (DMEM) supplemented with 10% fetal bovine serum and 1% penicillin/streptomycin mixed at 37 °C with 5% CO_2_. When the cell density reached approximately 80%, the cells were digested with trypsin for use in experiments.

### 4.8. Cell Proliferation

The CCK-8 assay was used to evaluate cell viability. H9c2 cells were seeded in a 96-well plate (5 × 10^4^ cells per well). After 24 h, different concentrations of ginsenoside Rb_1_ (0, 6.25, 12.5, 25, 50, 100, 200, and 400 μM) [15] were added to the experimental groups, while the positive control group was treated with the cardiotoxic drug doxorubicin (1 µM; 23214-92-8, Macklin, Shanghai, China). After exposure for 24 h or 48 h, cells were incubated with 10 μL of CCK-8 solution (C0038, Beyotime, Shanghai, China) and maintained at 37 °C with 5% CO_2_ for 1.5 h. The absorbance at OD = 450 nm was detected using a microplate reader (Silicon Valley, CA, USA).

### 4.9. Detection of Mitochondrial Structure and Function

#### 4.9.1. Ultrastructure of Myocardial Mitochondria Was Observed by Transmission Electron Microscopy

Heart tissues from the SD rats were collected and quickly placed into an electron microscope fixative solution at 4 °C for 2–4 h. After washing with PBS, the hearts were fixed with 1% osmic acid at room temperature, subjected to gradient dehydration for 2 h, embedded on tissue sections, stained with 2% uranyl acetate, and immersed in lead citrate for 15 min. Images were captured using a transmission electron microscope (JEOL, Tokyo, Japan).

#### 4.9.2. Changes in Mitochondrial Membrane Potential

MMP was assessed using the 5,5′,6,6′-Tetrachloro-1,1′,3,3′-tetraethyl-imidacarbocyanine iodide (JC-1) assay (C2005, Beyotime, Shanghai, China) following the manufacturer’s instructions. Myocardial tissue mitochondria was extracted; JC-1 was diluted with JC-1 staining buffer and was repeatedly blown and mixed with a pipette to obtain JC-1 staining working solution. In total, 0.1 mL of purified mitochondria with a total protein content of 100 μg was added to 0.9 mL of JC-1 staining solution. After mixing, it was directly detected using a fluorescence enzyme-labeled instrument. The green fluorescence (Ex = 514 m, Em = 529 nm) was used for monomer detection and the red fluorescence (Ex = 585 nm, Em = 590 nm) was used for J-aggregate detection.

After digestion and centrifugation, H9c2 cells were counted and seeded (5 × 10^4^ cells/mL), with 1 mL of cell suspension being inoculated into each well of a confocal dish. After 24 h, ET-1 (E7920, Solarbio, Beijing, China) was used to induce modeling [45]. Following 24 h of modeling, ginsenoside Rb_1_ (0, 6.25, 12.5, 25, 50, 100, 200, and 400 μM) was administered. After 24 h of drug administration, the complete medium was mixed with JC-1 staining solution (1:1), and the cells were incubated at 37 °C for 20 min. Subsequently, the cells were rinsed twice with ice-cold PBS and replaced with normal complete medium. Imaging was then performed under a confocal microscope.

#### 4.9.3. Detection of the Productivity Efficiency of ETC

The heart tissue was weighed, sliced, and then added to an NAD^+^/nicotinamide adenine dinucleotide (NADH) extraction buffer (S0175, Beyotime, China). The mixture was homogenized in an ice bath, centrifuged at 4 °C for 5 min, and the supernatant was centrifuged. A gradient dilution of 10 mM NADH standard with NAD^+^/NADH extraction buffer was performed, as well as a dilution of alcohol dehydrogenase working solution (90 μL/ detection hole) according to the detection requirements; the samples were measured according to the instructions. Finally, the ratio of NAD^+^/NADH in the sample was calculated.

The cell culture medium was sucked up; NAD^+^/NADH extract was added to the cells, which were gently blown to promote cell lysis, centrifuged at 4 °C for 5 min at 12,000× *g*, and the supernatant was taken to be tested.

#### 4.9.4. ATP Level Detection

After weighing the heart tissue, ATP lysate (S0027, Beyotime, China) was added for homogenization, centrifuged at 4 °C and 12,000× *g* for 5 min, and the supernatant was taken to be tested. A gradient dilution was carried out on the standard to several concentrations of 0.01, 0.05, 0.1, 0.5, 1, 5 and 10 μM. According to the number of samples to be tested, the ATP detection working solution was prepared and the ATP concentration was determined.

According to the ratio of 1/10 of cell culture solution, ATP lysis solution was added to myocardial cells to lyse the cells, which were centrifuged at 4 °C, 12,000× *g* for 5 min, and the supernatant was taken to be measured.

### 4.10. Western Blotting

Proteins were extracted from cardiac tissue samples. The cardiac tissue was washed with pre-cooled PBS and dissolved using RIPA lysis buffer. The tissue was lysed on ice for 30 min. Then, it was centrifuged at 4 °C, 12,000× *g* for 5 min. The supernatant was taken, the protein concentration was detected with a trace BCA protein detection kit, and loading buffer was added to the sample. A 10% gel was prepared and electrophoresed at 80 V for 30 min and 120 V for 1 h. The PVDF membrane was immersed in methanol for 15 s, and the separated protein was moved to the PVDF membrane at 300 mA for 1 h. The membrane was blocked with 5% skim milk for 1 h. The membrane was washed with TBST three times, primary antibody was added, and it was incubated at 4 °C overnight. Anti-*β*-actin antibody (81115-1-RR, 1:5000, Proteintech, China), anti-GLUT4 antibody (66846-1-1g, 1:2000, Proteintech, China), anti-PPARγ antibody (YT3836, 1:500, Immunoway, Suzhou, China), anti-Twist1 antibody (A25134, 1:1000, ABclonal, China), anti-PGC-1α antibody (ab313559, 1:1000, Abcam, Cambridge, UK), and anti-PPARα antibody (66826-1-1g, 1:1000, Proteintech, China) were used in the experiment. The PVDF membrane was subsequently incubated with horseradish peroxidase (HRP)-conjugated secondary antibodies, including anti-mouse IgG (SA00001-1, Proteintech, China) or anti-rabbit IgG (SA00001-2, Proteintech, China). Then, the PVDF membrane was washed with TBST three times. ECL chemiluminescence liquid (P10100, NCM, Beijing, China) was used for exposure. The gray values of the results were analyzed using ImageJ 1.5.3 software.

### 4.11. Molecular Docking of Ginsenoside Rb_1_ with Key Targets

Using molecular docking methods to verify the binding effect between small molecule drugs and potential targets can provide insights into the potential therapeutic effects of small molecule drugs on diseases. The Twist1 crystal structure was obtained from the RCSB-PDB Protein Data Bank (https://www.rcsb.org/) (accessed on 2 March 2025), and the active binding site of Twist1 (PDB code: 8OSB) was determined according to the literature [47]. The unrelated ligands of protein species were removed by PyMOL 3.1 software (https://www.pymol.org/) (accessed on 2 March 2025), and the protein structure in PDB format was derived. The molecular structure of ginsenoside Rb_1_ was downloaded from PubChem database and converted to PDB format using Open Babel 3.1.1 software. Hydrogenation and structural optimization were performed with AutoDock 1.5.7 software (https://www.pymol.org/) (accessed on 2 March 2025), followed by output conversion to PDBQT format. A molecular docking active pocket was designed to fully encompass the protein chain, integrating the original ligand position with the previously reported active binding site of Twist1 protein derived from literature. The molecular docking of Twist1 and ginsenoside Rb_1_ was carried out by using AutoDock 1.5.7 software. By calculating the affinity energy (kcal/mol), the binding stability of compound to receptor target proteins was evaluated. The affinity energy of <−5.0 kcal/mol was considered as a robust binding interaction between the compound and receptor protein. A lower negative affinity energy indicates a stronger affinity between the active compound and the target protein, resulting in a more stable structure. Finally, the docking results were imported into PyMOL 3.1 software for analysis and visualization.

### 4.12. ELISA Measurement

The concentrations of serum ANP, BNP (H180-1-2, H166-1-2, NJCB, Nanjing, China), and cTnT (E-EL-R0151, Elabscience, Wuhan, China) in the rats were measured using ELISA kits. Absorbance at OD = 450 nm was measured using a microplate reader.

RIPA lysate was added to the treated cells, which were then centrifuged and collected from the supernatant to detect the levels of Twist1, PGC-1α (m1690311, m1037076, mlbio, China), and PPARα (F3129-A, FANKEW, Chengdu, China) in the cells.

### 4.13. Statistical Analysis

Quantitative data are presented as mean ± standard error of the mean (Mean ± SEM). All the studies were designed to generate groups of equal size, using randomization and blinded analysis. Statistical analysis was performed using the GraphPad Prism 8.0 (GraphPad Sofeware 8.0.2). Tukey’s multiple comparison test was used after one-way analysis of variance (ANOVA) for multiple groups. Statistical significance was defined as *p* < 0.05.

## 5. Conclusions

In summary, this study confirmed that ginsenoside Rb_1_ is a compound that inhibits ventricular remodeling in HF by improving energy metabolism disorder in HF. Ginsenoside Rb_1_ has the effect of maintaining myocardial mitochondrial energy metabolism in rats with HF, reducing myocardial fibrosis, and thus improving cardiac function in rats. The mechanism of ginsenoside Rb_1_ in treating HF is achieved by regulating the Twist1/PGC-1α/PPARα signaling pathway. This study clarifies the molecular mechanism of ginsenoside Rb_1_ in inhibiting HF, and suggests that Twist1 may become a clinical diagnostic marker of HF. Twist1 expression levels reflect HF-related myocardial fibrosis and therapeutic response. As both a predictive biomarker and therapeutic target, Twist1 provides critical support for monitoring treatment efficacy, and implementing precision antifibrotic interventions in HF management.

## Figures and Tables

**Figure 1 pharmaceuticals-18-00500-f001:**
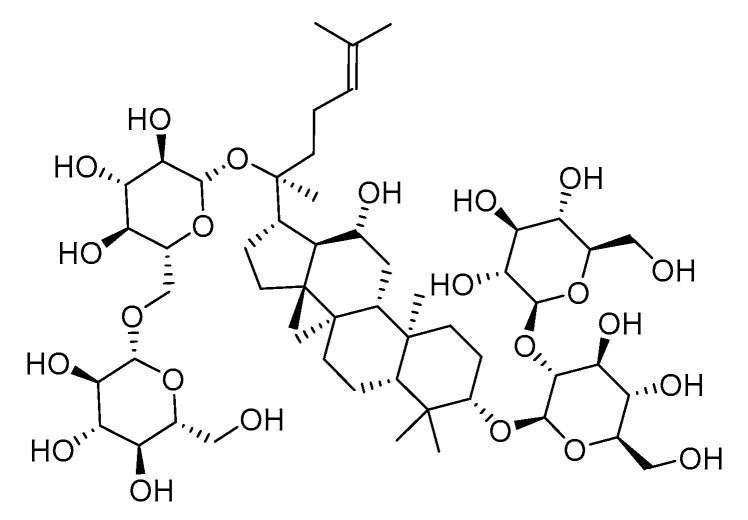
Chemical structure of ginsenoside Rb_1_.

**Figure 2 pharmaceuticals-18-00500-f002:**
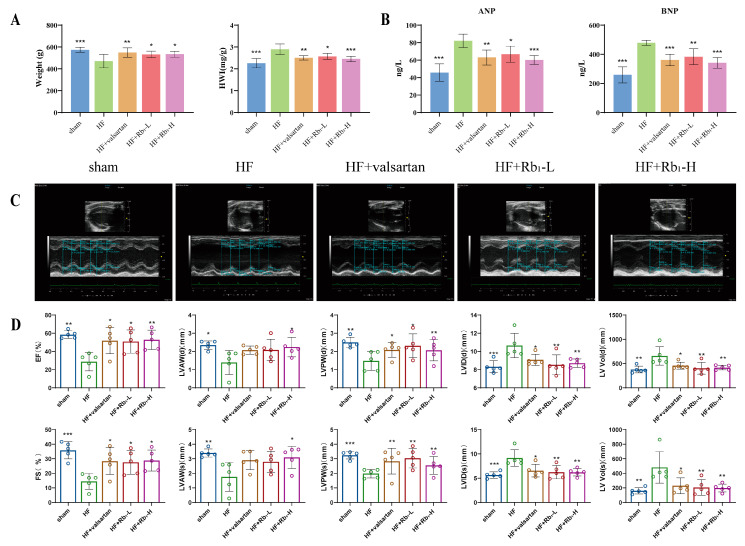
Ginsenoside Rb_1_ improves cardiac function and myocardial injury in rats with HF. (**A**) Weight and HWI of rats with HF. (**B**) Measurement of serum biochemical indicators ANP and BNP (*n* = 6). (**C**) Representative M-mode images of echocardiography. (**D**) Echocardiographic parameters of EF, FS, LVAW(d), LVAW(s), LVPW(d), LVPW(s), LVID(d), LVID(s), LV Vol(d), and LV Vol(s) (*n* = 5). Compared with the HF group, *** *p* < 0.001, ** *p* < 0.01, * *p* < 0.05. Sham: the sham group; HF: HF model group; HF + valsartan: valsartan treatment group (8 mg/kg); HF + Rb_1_-L: Rb_1_ treatment group (40 mg/kg); HF + Rb_1_-H: Rb_1_ treatment group (80 mg/kg).

**Figure 3 pharmaceuticals-18-00500-f003:**
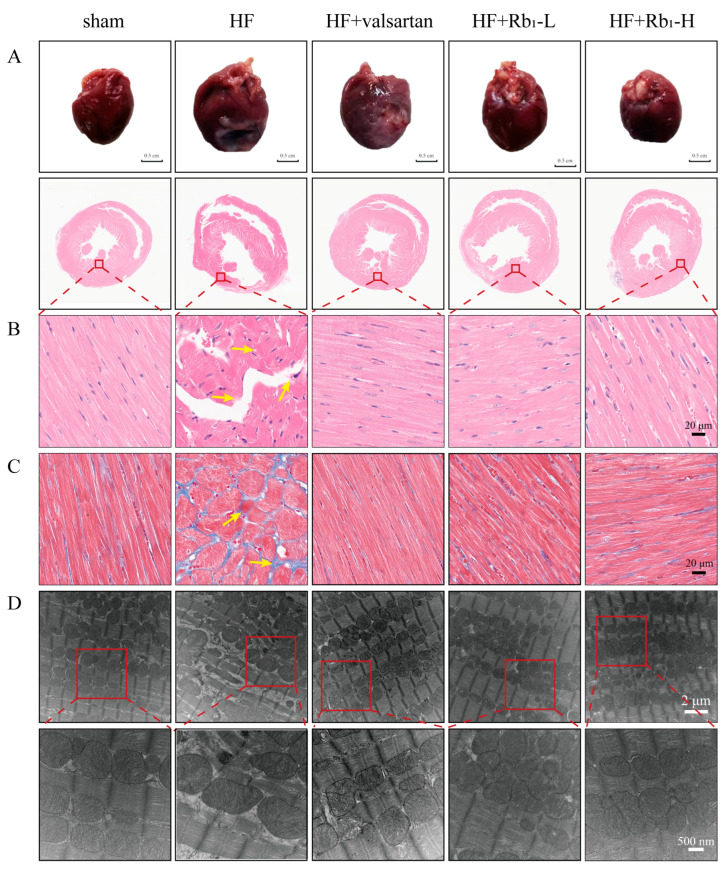
Ginsenoside Rb_1_ can improve the pathological changes of the left ventricle in rats with HF. (**A**) Pictures of rat hearts. (**B**) Representative HE-stained myocardial tissue sections (*n* = 6). Bar = 20 µm (40× original magnification). The yellow arrow indicates the typical pathological changes of myocardial tissue. (**C**) Representative Masson-stained images of cardiac tissue sections (*n* = 6). Bar = 20 µm (40× original magnification). The yellow arrow indicates the typical pathological changes of myocardial tissue. (**D**) Observation on the ultrastructure of the hearts of rats with HF screened by a transmission electron microscope (*n* = 5). Bar = 2 µm or Bar = 500 nm (10,000× original magnification or 30,000× original magnification, the picture below the red dotted line is an enlargement of the picture above.).

**Figure 4 pharmaceuticals-18-00500-f004:**
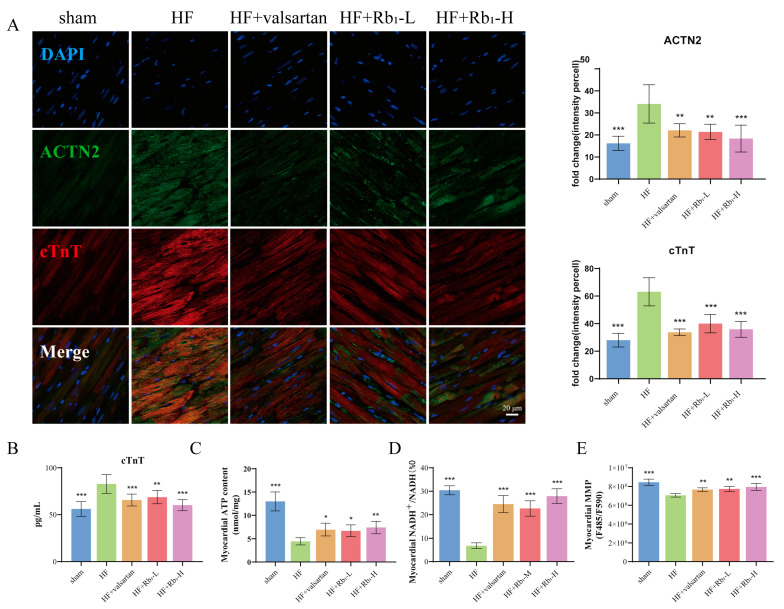
Ginsenoside Rb_1_ can improve myocardial injury in rats with HF. (**A**) Evaluation of cTnT and ACTN2 expression of cardiac tissues on rats with HF using immunofluorescence. (*n* = 6). Bar = 20 µm (40× original magnification). (**B**) Evaluation of cTnT expression in the serum of rats with HF using ELISA method (*n* = 6). (**C**–**E**) Ginsenoside Rb_1_ regulated ATP content, ETC productivity efficiency, and MMP level in the heart tissue of rats with HF (*n* = 6). Compared with the HF group, *** *p* < 0.001, ** *p* < 0.01, * *p* < 0.05.

**Figure 5 pharmaceuticals-18-00500-f005:**
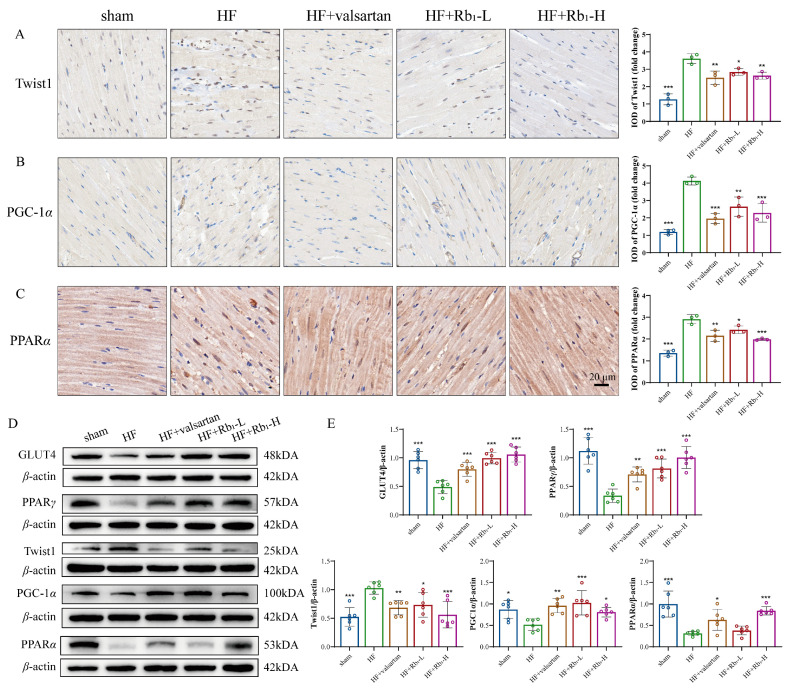
(**A**,**B**) Evaluation of GLUT4, PPARγ, Twist1, PGC-1α, and PPARα expression on rats with HF with a quantification of the average relative density of WB data. (**C**–**E**) Evaluation of Twist1, PGC-1α, and PPARα expression of cardiac tissues on rats with HF by immunohistochemistry. Compared with the HF group, *** *p* < 0.001, ** *p* < 0.01, * *p* < 0.05.

**Figure 6 pharmaceuticals-18-00500-f006:**
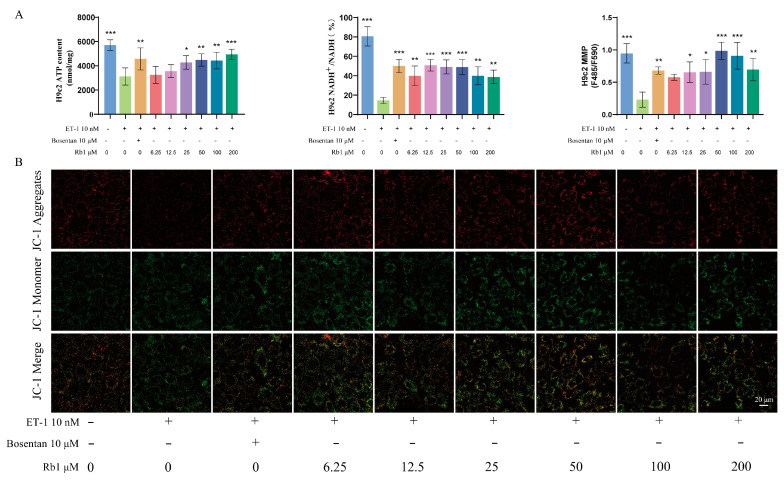
Detection of energy metabolism indexes of H9c2 cells: (**A**) Ginsenoside Rb_1_ regulated ATP content (*n* = 6), ETC productivity efficiency (*n* = 3), and MMP level (*n* = 3). (**B**) JC-1 mitochondrial membrane potential staining results. Compared with the ET-1 model group, *** *p* < 0.001, ** *p* < 0.01, * *p* < 0.05.

**Figure 7 pharmaceuticals-18-00500-f007:**
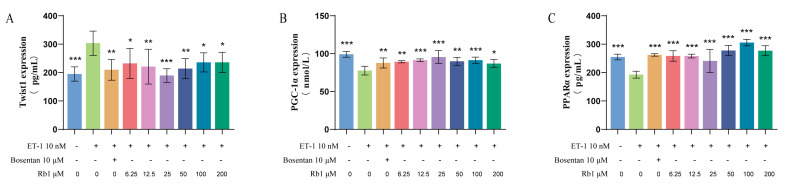
Expression of energy metabolism protein regulated by ginsenoside Rb_1_ in H9c2 cells was detected by ELISA: (**A**) Twist1, (**B**) PGC-1α, (**C**) PPARα (*n* = 6). Compared with the ET-1 model group, *** *p* < 0.001, ** *p* < 0.01, * *p* < 0.05.

**Figure 8 pharmaceuticals-18-00500-f008:**
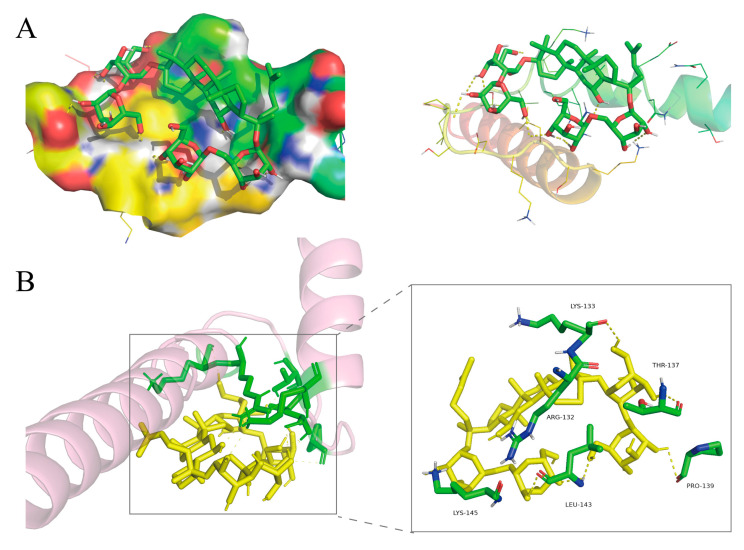
Ginsenoside Rb_1_ and Twist1 (**A**) molecular docking results, (**B**) amino acid residues interaction.

**Figure 9 pharmaceuticals-18-00500-f009:**
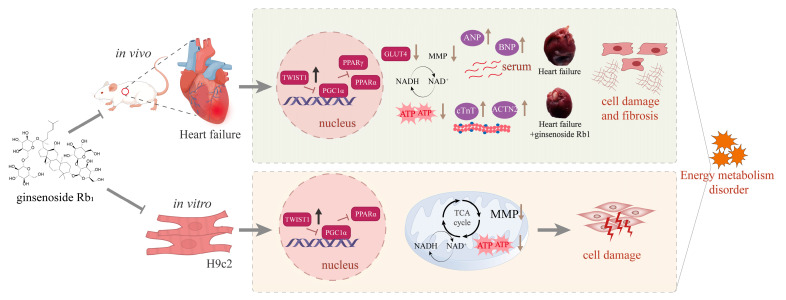
Ginsenoside Rb_1_ alleviates ventricular remodeling in HF by regulating the Twist1/PGC-1α/PPARα signaling pathway. In the hearts of rats with HF, there is an increase in the expression of cTnT and ACTN2. Additionally, there is an elevation in atrial and ventricular pressures, accompanied by an increased secretion of ANP and BNP. Myocardial cell damage occurs, inhibiting glucose transport (reduced GLUT4 synthesis), leading to mitochondrial dysfunction. This dysfunction manifests as a decrease in MMP, reduced ATP synthesis, and decreased ETC productivity efficiency, resulting in disordered energy metabolism. Furthermore, an abnormally high expression of Twist1 is observed, which inhibits the expression of PGC-1α, PPARα, and PPARγ, thereby exacerbating cardiac injury.

## Data Availability

The data that support the findings of this study are available from the corresponding author upon reasonable request. The data are not publicly available due to [confidentiality obligations associated with the ongoing research project].

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
