# Peer review of "Ginsenoside Rb1 Ameliorates Heart Failure Ventricular Remodeling by Regulating the Twist1/PGC-1α/PPARα Signaling Pathway"

_pharmaceuticals, 2025, doi:10.3390/ph18040500_

Round 1
Reviewer 1 Report
Comments and Suggestions for Authors
The article is a study seeking to show that ginsenoside Rb1 inhibits ventricular remodeling in HF by improving the energy metabolism disorder in HF. The methodology implemented by the authors leads to results confirming this, particularly the mechanism of action.
Ginsenoside Rb1 is compared to valsartan, which does not contain any carbohydrate parts. Would it not be interesting to also compare ginsenoside Rb1 with its aglycone part? Is the carbohydrate part as important in the action of the molecule?
The authors should better highlight the perspectives of their studies, and thus describe in a few sentences the real applied interest of the confirmation obtained.
Author Response
Dear editor and reviewer,
We are very grateful to the reviewers for their willingness to spend time providing constructive comments. These comments are very valuable and helpful, and we have made point-to-point modifications based on relevant feedback.
Comments 1: [Ginsenoside Rb1 is compared to valsartan, which does not contain any carbohydrate parts. Would it not be interesting to also compare ginsenoside Rb1 with its aglycone part? Is the carbohydrate part as important in the action of the molecule?]
Response 1: Thank you for pointing this out. We have explained your questions as follows. [The purpose of this study is to explore the effect and mechanism of ginsenoside Rb1 on reducing ventricular remodeling in HF, so valsartan was selected as a positive control drug to observe the effect of ginsenoside Rb1 more objectively. Valsartan is a first-line drug in clinical practice. Its cardioprotective effect has been confirmed in large-scale clinical trials. These trials have shown that after taking valsartan, the incidence rate of cardiovascular morbidity and mortality in patients with HF after myocardial infarction, as well as patients with hypertension, coronary heart disease, and/or HF, significantly decreased, and the hospitalization rate of HF also decreased (Black et al., 2009). In addition, valsartan improved cardiac structure and function by reducing ventricular hypertrophy and myocardial fibrosis (Ho et al., 2021). Therefore, we choose valsartan as a positive drug.
The aglycone part of ginsenoside Rb1 is protopanaxadiol (PPD), which contains no glycosyl groups, while the carbohydrate part consists of glycosyl groups such as glucose and arabinose. PPD is the core active ingredient of ginsenoside Rb1. The exposure degree of PPD directly determines the efficacy of metabolites. Compared with ginsenoside Rb1, PPD exhibits enhanced lipophilicity, improved intestinal absorption, increased bioactivity and pharmacological effects. However, the glycosyl group in ginsenoside Rb1 is equally critical, as it balances the water solubility and dissolution properties of the compound, thereby improving its bioavailability(Kim et al., 2013) (Akao et al., 1998; Bai and Gänzle, 2015; Lee et al., 2005; Shen et al., 2018; Shen et al., 2013). Whether metabolites retaining a glycosyl group post-metabolism can deliver energy to cardiomyocytes upon cardiac entry remains to be elucidated.]
References:
Black, H. R., Bailey, J., Zappe, D., Samuel, R., 2009. Valsartan: more than a decade of experience. Drugs, 69(17): 2393-2414.
Ho, C. Y., Day, S. M., Axelsson, A., Russell, M. W., Zahka, K., Lever, H. M., Pereira, A. C., Colan, S. D., Margossian, R., Murphy, A. M., Canter, C., Bach, R. G., Wheeler, M. T., Rossano, J. W., Owens, A. T., Bundgaard, H., Benson, L., Mestroni, L., Taylor, M. R. G., Patel, A. R., Wilmot, I., Thrush, P., Vargas, J. D., Soslow, J. H., Becker, J. R., Seidman, C. E., Lakdawala, N. K., Cirino, A. L., Burns, K. M., McMurray, J. J. V., MacRae, C. A., Solomon, S. D., Orav, E. J., Braunwald, E., 2021. Valsartan in early-stage hypertrophic cardiomyopathy: a randomized phase 2 trial. Nat Med, 27(10): 1818-1824.
Kim, E.-M., Seo, J.-H., Kim, J., Park, J.-S., Kim, D.-H., Kim, B.-G., 2013. Production of ginsenoside aglycons and Rb1 deglycosylation pathway profiling by HPLC and ESI-MS/MS using Sphingobacterium multivorum GIN723. Appl Microbiol Biotechnol, 97(18): 8031-8039.
Akao, T., Kida, H., Kanaoka, M., Hattori, M., Kobashi, K., 1998. Intestinal bacterial hydrolysis is required for the appearance of compound K in rat plasma after oral administration of ginsenoside Rb1 from Panax ginseng. J Pharm Pharmacol, 50(10): 1155-1160.
Bai, Y., Gänzle, M. G., 2015. Conversion of ginsenosides by Lactobacillus plantarum studied by liquid chromatography coupled to quadrupole trap mass spectrometry. Food Res Int, 76(Pt 3): 709-718.
Lee, H.-U., Bae, E.-A., Han, M. J., Kim, N.-J., Kim, D.-H., 2005. Hepatoprotective effect of ginsenoside Rb1 and compound K on tert-butyl hydroperoxide-induced liver injury. Liver Int, 25(5): 1069-1073.
Shen, H., Gao, X.-J., Li, T., Jing, W.-H., Han, B.-L., Jia, Y.-M., Hu, N., Yan, Z.-X., Li, S.-L., Yan, R., 2018. Ginseng polysaccharides enhanced ginsenoside Rb1 and microbial metabolites exposure through enhancing intestinal absorption and affecting gut microbial metabolism. J Ethnopharmacol, 216: 47-56.
Shen, H., Leung, W.-I., Ruan, J.-Q., Li, S.-L., Lei, J. P.-C., Wang, Y.-T., Yan, R., 2013. Biotransformation of ginsenoside Rb1 via the gypenoside pathway by human gut bacteria. Chin Med, 8(1): 22.
Comments 2: [The authors should better highlight the perspectives of their studies, and thus describe in a few sentences the real applied interest of the confirmation obtained.]
Response 2: Agreed. We have revised the manuscript as following in (page 3, paragraph 1, line 24-29). [Ginsenoside Rb1 can inhibit the upregulation of Twist1 and activate the expression of its downstream PGC-1α and PPARα expression. By modulating the Twist1/PGC-1α/PPARα signaling pathway, alleviating ventricular remodeling in HF patients and improving myocardial energy metabolism dysfunction. Twist1 may be a key target for the treatment of HF. This study not only elucidates the mechanism by which ginsenoside Rb1 alleviates HF, but also provides new insights into the clinical treatment of HF.]

Reviewer 2 Report
Comments and Suggestions for Authors
- Language is fine.
- Abstract: is well written.
- Introduction: please add information about mitochondrial dysfunction and Ginsenoside Rb1 relation.
- Results: please reevaluate the figure 2 statistical results for the treatment groups.
- Please add arrows to section B and C. What do we have to see in the images?
- Discussion: well written.
Author Response
Dear editor and reviewer,
We are very grateful to the reviewers for their willingness to spend time providing constructive comments. These comments are very valuable and helpful, and we have made point-to-point modifications based on relevant feedback.
Comments 1: [Introduction: please add information about mitochondrial dysfunction and Ginsenoside Rb1 relation.]
Response 1: At your prompt, we revised the manuscript in (page 5, paragraph 2, lines 3-17; page 6, paragraph 2) [Ginsenoside Rb1, a major active component of Panax ginseng, exhibits broad pharmacological effects. In previous studies in rat models of myocardial infarction (MI)-induced HF, ginsenoside Rb1 restored cardiac function and alleviated HF through multi-pathway and multi-target mechanisms, including mitigating oxidative stress, exerting anti-inflammatory effects, and improving mitochondrial quality control. Du Lixin et al. demonstrated that ginsenoside Rb1- PLGA nanoparticles significantly improved myocardial oxidative stress damage and pathological conditions in HF rats by activating the reactive oxygen species (ROS)/ peroxisome proliferator-activated receptor alpha (PPARα)/ peroxisome proliferator-activated receptor gamma coactivator 1α (PGC-1α) pathway (Du et al., 2023). Meanwhile, Zheng Zhi's team developed a biomimetic adhesive- injectable hydrogel targeting mitochondrial DNA (mtDNA)-STING signaling crosstalk, which enhanced inflammation clearance and mitochondrial repair to promote MI recovery(Zheng et al., 2024). Notably, ginsenoside Rb1 further improved mitochondrial quality control by regulating the dual-specificity phosphatase-1 (DUSP-1)- BAX inhibitor motif containing 6 (TMBIM6) axis, suppressed inflammatory responses, and modulated gut microbiota to maintain mitochondrial homeostasis, ultimately reversing HF progression (Pu et al., 2024).
Additionally, ginsenoside Rb1 effectively improved cardiac dysfunction in HF mice and isoproterenol-induced HF rats, reducing myocardial hypertrophy, fibrosis, and collagen deposition. It restored energy metabolism by regulating cardiac enzyme activity and fatty acid oxidation, promoting ATP production, maintaining mitochondrial function, and delaying HF progression(Ke et al., 2020; Li et al., 2023). However, the molecular mechanisms by which ginsenoside Rb1 inhibits cardiac fibrosis in MI-induced HF rats via improving energy metabolism remain unclear.]
References:
Du, L., Lu, H., Wang, Z., Liu, C., Xiao, Y., Guo, Z., Li, Y., 2023. Therapeutic Potential of Ginsenoside Rb1-PLGA Nanoparticles for Heart Failure Treatment via the ROS/PPARα/PGC1α Pathway. Molecules, 28(24).
Zheng, Z., Sun, J., Wang, J., He, S., Liu, Z., Xie, J., Yu, C.-Y., Wei, H., 2024. Enhancing myocardial infarction treatment through bionic hydrogel-mediated spatial combination therapy via mtDNA-STING crosstalk modulation. J Control Release, 371: 570-587.
Pu, X., Zhang, Q., Liu, J., Wang, Y., Guan, X., Wu, Q., Liu, Z., Liu, R., Chang, X., 2024. Ginsenoside Rb1 ameliorates heart failure through DUSP-1-TMBIM-6-mediated mitochondrial quality control and gut flora interactions. Phytomedicine, 132: 155880.
Ke, S.-Y., Liu, D.-H., Wu, L., Yu, X.-G., Wang, M., Shi, G.-Y., Wen, R.-H., Zhou, B., Hao, B.-S., Liu, Y., Zhu, J.-M., Qian, X.-X., 2020. Ginsenoside Rb1 Ameliorates Age-Related Myocardial Dysfunction by Regulating the NF- κB Signaling Pathway. Am J Chin Med, 48(6): 1369-1383.
Li, C., Zhang, X., Li, J., Liang, L., Zeng, J., Wen, M., Pan, L., Lv, D., Liu, M., Cheng, Y., Huang, H., 2023. Ginsenoside Rb1 promotes the activation of PPARα pathway via inhibiting FADD to ameliorate heart failure. Eur J Pharmacol, 947: 175676.
Comments 2: [ Results: please reevaluate the figure 2 statistical results for the treatment groups. Please add arrows to section B and C. What do we have to see in the images?]
Response 2: Thank you for pointing out the modification. [We have modified this figure and indicated the typical pathological changes with yellow arrows.]

Reviewer 3 Report
Comments and Suggestions for Authors
- The abstract should include a concluding statement highlighting the key findings and their clinical implications.
- Similarly, in the introduction section, the authors should clearly state the research gap identified from the literature. Subsequently, a discussion should be strengthened, and the significance of their study should be emphasized in detail.
- Further, a thorough discussion with additional references on similar studies that employed ginsenoside Rb1 in heart failure models should be strengthened.
- It is not clear in the introduction section why the authors specifically chose Twist1/PGC-1α/PPARα pathway for this study.
- In the methodology section, the statistical analyses employed in this study should be discussed in-depth.
- The authors should include more details on how the molecular docking study was conducted. They should include the details of the specific software parameters.
- The authors should clarify the rationale behind selecting specific dosages and time points in the methodology section.
- The authors should ensure that all figure labels and legends are large enough. The labels and legends in many graphs are not legible at 100% magnification.
- In the discussion, the authors have successfully connected findings to prior research. However, I would like them to strengthen the discussion to emphasize the clinical implications of the results.
- Though the authors have acknowledged the limitations of the study, further discussion on the prospective future research directions based on the limitations should be added.
- In the conclusion section, the authors should reinforce the potential applications of their outcome in clinical settings contributions more explicitly.
The English could be improved to more clearly express the research.
Author Response
Dear editor and reviewer,
We are very grateful to the reviewers for their willingness to spend time providing constructive comments. These comments are very valuable and helpful, and we have made point-to-point modifications based on relevant feedback.
Comments 1: [The abstract should include a concluding statement highlighting the key findings and their clinical implications.]
Response 1: Agreed. We have revised the manuscript as following in (page 3, paragraph 1, line 24-29). [Ginsenoside Rb1 can inhibit the upregulation of Twist1 and activate the expression of its downstream PGC-1α and PPARα expression. By modulating the Twist1/PGC-1α/PPARα signaling pathway, alleviating ventricular remodeling in HF patients and improving myocardial energy metabolism dysfunction. Twist1 may be a key target for the treatment of HF. This study not only elucidates the mechanism by which ginsenoside Rb1 alleviates HF, but also provides new insights into the clinical treatment of HF.]
Comments 2: [Similarly, in the introduction section, the authors should clearly state the research gap identified from the literature. Subsequently, a discussion should be strengthened, and the significance of their study should be emphasized in detail.]
Response 2: Thank you for pointing out the modification. At your prompt, we revised the manuscript in (page 5, paragraph 1, lines 9-17). [Mitochondrial energy damage and energy metabolism disorder are important characteristics of HF, which lead to insufficient myocardial productivity and induce ventricular remodeling in HF (Zhou and Tian, 2018). Despite groundbreaking advances in guideline-directed medical therapy and remarkable improvements in therapeutic outcomes, the epidemiological burden of HF continues to escalate, with persistently rising incidence and mortality rates globally. Current therapies targeting HF-related energy metabolism disorders face challenges such as unclear mechanisms of action and oversimplified intervention strategies. Their limited efficacy leaves most patients do not receive optimal treatment (Dedkova et al., 2013; Morales et al., 2020; Mouquet et al., 2010). Chinese medicine represented by ginseng has the function of regulating cardiac energy metabolism, which may be a safer and more effective treatment for HF.]
References:
Zhou, B., Tian, R., 2018. Mitochondrial dysfunction in pathophysiology of heart failure. J Clin Invest, 128(9): 3716-3726.
Dedkova, E. N., Seidlmayer, L. K., Blatter, L. A., 2013. Mitochondria-mediated cardioprotection by trimetazidine in rabbit heart failure. J Mol Cell Cardiol, 59: 41-54.
Morales, P. E., Arias-Durán, C., Ávalos-Guajardo, Y., Aedo, G., Verdejo, H. E., Parra, V., Lavandero, S., 2020. Emerging role of mitophagy in cardiovascular physiology and pathology. Mol Aspects Med, 71: 100822.
Mouquet, F., Rousseau, D., Domergue-Dupont, V., Grynberg, A., Liao, R., 2010. Effects of trimetazidine, a partial inhibitor of fatty acid oxidation, on ventricular function and survival after myocardial infarction and reperfusion in the rat. Fundam Clin Pharmacol, 24(4): 469-476.
Comments 3: [Further, a thorough discussion with additional references on similar studies that employed ginsenoside Rb1 in heart failure models should be strengthened.]
Response 3: We have revised the manuscript as following in (page 5, paragraph 2, line 3-17; page 6, paragraph 2). [Ginsenoside Rb1, a major active component of Panax ginseng, exhibits broad pharmacological effects. In previous studies in rat models of myocardial infarction (MI)-induced HF, ginsenoside Rb1 restored cardiac function and alleviated HF through multi-pathway and multi-target mechanisms, including mitigating oxidative stress, exerting anti-inflammatory effects, and improving mitochondrial quality control. Du Lixin et al. demonstrated that ginsenoside Rb1- PLGA nanoparticles significantly improved myocardial oxidative stress damage and pathological conditions in HF rats by activating the reactive oxygen species (ROS)/ peroxisome proliferator-activated receptor alpha (PPARα)/ peroxisome proliferator-activated receptor gamma coactivator 1α (PGC-1α) pathway (Du et al., 2023). Meanwhile, Zheng Zhi's team developed a biomimetic adhesive- injectable hydrogel targeting mitochondrial DNA (mtDNA)-STING signaling crosstalk, which enhanced inflammation clearance and mitochondrial repair to promote MI recovery(Zheng et al., 2024). Notably, ginsenoside Rb1 further improved mitochondrial quality control by regulating the dual-specificity phosphatase-1 (DUSP-1)- BAX inhibitor motif containing 6 (TMBIM6) axis, suppressed inflammatory responses, and modulated gut microbiota to maintain mitochondrial homeostasis, ultimately reversing HF progression (Pu et al., 2024).
Additionally, ginsenoside Rb1 effectively improved cardiac dysfunction in HF mice and isoproterenol-induced HF rats, reducing myocardial hypertrophy, fibrosis, and collagen deposition. It restored energy metabolism by regulating cardiac enzyme activity and fatty acid oxidation, promoting ATP production, maintaining mitochondrial function, and delaying HF progression(Ke et al., 2020; Li et al., 2023). However, the molecular mechanisms by which ginsenoside Rb1 inhibits cardiac fibrosis in MI-induced HF rats via improving energy metabolism remain unclear.]
References:
Du, L., Lu, H., Wang, Z., Liu, C., Xiao, Y., Guo, Z., Li, Y., 2023. Therapeutic Potential of Ginsenoside Rb1-PLGA Nanoparticles for Heart Failure Treatment via the ROS/PPARα/PGC1α Pathway. Molecules, 28(24).
Zheng, Z., Sun, J., Wang, J., He, S., Liu, Z., Xie, J., Yu, C.-Y., Wei, H., 2024. Enhancing myocardial infarction treatment through bionic hydrogel-mediated spatial combination therapy via mtDNA-STING crosstalk modulation. J Control Release, 371: 570-587.
Pu, X., Zhang, Q., Liu, J., Wang, Y., Guan, X., Wu, Q., Liu, Z., Liu, R., Chang, X., 2024. Ginsenoside Rb1 ameliorates heart failure through DUSP-1-TMBIM-6-mediated mitochondrial quality control and gut flora interactions. Phytomedicine, 132: 155880.
Ke, S.-Y., Liu, D.-H., Wu, L., Yu, X.-G., Wang, M., Shi, G.-Y., Wen, R.-H., Zhou, B., Hao, B.-S., Liu, Y., Zhu, J.-M., Qian, X.-X., 2020. Ginsenoside Rb1 Ameliorates Age-Related Myocardial Dysfunction by Regulating the NF-κB Signaling Pathway. Am J Chin Med, 48(6): 1369-1383.
Li, C., Zhang, X., Li, J., Liang, L., Zeng, J., Wen, M., Pan, L., Lv, D., Liu, M., Cheng, Y., Huang, H., 2023. Ginsenoside Rb1 promotes the activation of PPARα pathway via inhibiting FADD to ameliorate heart failure. Eur J Pharmacol, 947: 175676.
Comments 4: [It is not clear in the introduction section why the authors specifically chose Twist1/PGC-1α/PPARα pathway for this study.]
Response 4: Thank you for pointing this out. We have explained your questions as following in (page 6, paragraph 3, line 3-14). [The relationship between energy metabolism disorders and ventricular remodeling in HF has been established in early research. PGC-1α and PPARα are recognized as key factors to control energy homeostasis, which affect myocardial energy metabolism by regulating mitochondrial biosynthesis, fatty acid oxidation and glucose metabolism (Mohan et al., 2023; Wu et al., 2024). The latest research shows that Twist1 is an upstream regulator of PGC-1α and PPARα, which plays a significant role in regulating energy metabolism. According to reports, Twist1 increases under chronic hypoxia and pathogenic conditions. Blocking the expression of Twist1 can alleviate mitochondrial dysfunction and intracellular lipid accumulation; promote the expression of PGC-1α and downstream target genes. It is worth noting that Twist1 has the function of regulating energy metabolism and inhibiting fibrosis, and can be used as a potential intervention target for anti-fibrosis treatment (Liu et al., 2022). However, how Twist1 plays a role in HF ventricular remodeling characterized by myocardial fibrosis has not been clarified.]
References:
Mohan, U. P., Pichiah, P. B. T., Arunachalam, S., 2023. Adriamycin downregulates the expression of KLF4 in cardiomyocytes in vitro and contributes to impaired cardiac energy metabolism in Adriamycin-induced cardiomyopathy. 3 Biotech, 13(5): 162.
Wu, C., Zhang, C., Li, F., Yan, Y., Wu, Y., Li, B., Tong, H., Lang, J., 2024. Fucoxanthin Mitigates High-Fat-Induced Lipid Deposition and Insulin Resistance in Skeletal Muscle through Inhibiting PKM1 Activity. J Agric Food Chem, 72(32): 18013-18026.
Liu, L., Ning, X., Wei, L., Zhou, Y., Zhao, L., Ma, F., Bai, M., Yang, X., Wang, D., Sun, S., 2022. Twist1 downregulation of PGC-1α decreases fatty acid oxidation in tubular epithelial cells, leading to kidney fibrosis. Theranostics, 12(8): 3758-3775.
Comments 5: [In the methodology section, the statistical analyses employed in this study should be discussed in-depth.]
Response 5: According to your request, we have modified the statistical analysis in (page 24, paragraph 4). [Quantitative data are presented as mean ± standard error of the mean (Mean ± SEM). All the studies were designed to generate groups of equal size, using randomization and blinded analysis. Statistical analysis was performed using the GraphPad Prism 8.0 (GraphPad Sofeware). Tukey’s multiple comparison test was used after one- way analysis of variance (ANOVA) for multiple groups. Statistical significance was defined as P < 0.05.]
Comments 6: [The authors should include more details on how the molecular docking study was conducted. They should include the details of the specific software parameters.]
Response 6: We have revised the manuscript as following in (page 23-24, paragraph 4). [Using molecular docking methods to verify the binding effect between small molecule drugs and potential targets can provide insights into the potential therapeutic effects of small molecule drugs on diseases. The Twist1 crystal structure was obtained from the RCSB-PDB Protein Data Bank (https://www.rcsb.org/), and the active binding site of Twist1 (PDB code: 8OSB) was determined according to the literature (Kim et al., 2024). The unrelated ligands of protein species were removed by PyMOL 3.1 software (https://www.pymol.org/), and the protein structure in PDB format was derived. The molecular structure of ginsenoside Rb1 was downloaded from PubChem database, and converted to PDB format using Open Babel 3.1.1 software. Hydrogenation and structural optimization were performed with AutoDock 1.5.7 software (https://www.pymol.org/), followed by output conversion to PDBQT format. A molecular docking active pocket was designed to fully encompass the protein chain, integrating the original ligand position with the previously reported active binding site of Twist1 protein derived from literature. The molecular docking of Twist1 and ginsenoside Rb1 was carried out by using AutoDock 1.5.7 software. By calculating the affinity energy (kcal/mol), the binding stability of compound to receptor target proteins was evaluated. The affinity energy of < −5.0 kcal/mol was considered as a robust binding interaction between the compound and receptor protein. A lower negative affinity energy indicates a stronger affinity between the active compound and the target protein, resulting in a more stable structure. Finally, the docking results were imported into PyMOL 3.1 software for analysis and visualization.]
References:
Kim, S., Morgunova, E., Naqvi, S., Goovaerts, S., Bader, M., Koska, M., Popov, A., Luong, C., Pogson, A., Swigut, T., Claes, P., Taipale, J., Wysocka, J., 2024. DNA-guided transcription factor cooperativity shapes face and limb mesenchyme. Cell, 187(3).
Comments 7: [The authors should clarify the rationale behind selecting specific dosages and time points in the methodology section.]
Response 7: At your prompt, we revised the manuscript in (page 19, paragraph 3, lines 3-8). [Based on the dosage and administration timing of ginsenoside Rb1 reported in previous literature for SD rats (Li et al., 2023), and combined with our research group's preliminary studies on the post-surgical model establishment and treatment time in rat with HF following MI (Wei et al., 2019), we determined the drug dosage and time points used in the current experiment. Four weeks post-surgery, the 32 rats with confirmed HF were randomly divided into 4 groups, and received daily oral gavage treatments for 6 weeks.]
References:
Li, C., Zhang, X., Li, J., Liang, L., Zeng, J., Wen, M., Pan, L., Lv, D., Liu, M., Cheng, Y., Huang, H., 2023. Ginsenoside Rb1 promotes the activation of PPARα pathway via inhibiting FADD to ameliorate heart failure. Eur J Pharmacol, 947: 175676.
Wei, J., Guo, F., Zhang, M., Xian, M., Wang, T., Gao, J., Wu, H., Song, L., Zhang, Y., Li, D., Yang, H., Huang, L., 2019. Signature-oriented investigation of the efficacy of multicomponent drugs against heart failure. FASEB J, 33(2): 2187-2198.
Comments 8: [The authors should ensure that all figure labels and legends are large enough. The labels and legends in many graphs are not legible at 100% magnification.]
Response 8: Thank you for pointing this out. [We have processed all figure labels and legends in the text to improve the clarity.]
Comments 9: [In the discussion, the authors have successfully connected findings to prior research. However, I would like them to strengthen the discussion to emphasize the clinical implications of the results.]
Response 9: Thank you for pointing out the modification. We have revised the manuscript as following in (page 17-18, paragraph 3, line 9-18). [Notably, our study revealed that Twist1 is highly expressed during ventricular remodeling in HF. This upregulation correlates with myocardial fibrosis. Therefore, monitoring Twist1 levels can help assess fibrosis severity and guide anti-fibrotic therapy. In addition, combining Twist1 with its downstream targets (PGC-1α and PPARα) improves diagnostic accuracy. As a sentinel molecule coordinating early remodeling events, Twist1 shows unique clinical value in monitoring therapeutic responses and guiding targeted interventions. Twist1, as a predictive biomarker and a novel therapeutic target, has broad application prospects in optimizing the treatment of HF.]
Comments 10: [Though the authors have acknowledged the limitations of the study, further discussion on the prospective future research directions based on the limitations should be added.]
Response 10: We have revised the manuscript as following in (page 18, paragraph 1, line 18-22). [Future studies should further validate the molecular mechanisms of Twist1 in MI-induced HF through Twist1 overexpression or knockout experiments. We will focus on investigating the regulatory effects of ginsenoside Rb1 on Twist1-mediated ventricular remodeling in HF, with a specific emphasis on elucidating its underlying mechanisms. The experimental design includes in vivo administration of ginsenoside Rb1 to cardiac-specific Twist1-overexpressing HF rat models, and in vitro treatment of Twist1-overexpressing cardiomyocytes with ginsenoside Rb1 to verify its regulatory role in HF pathogenesis. Based on these findings, ginsenoside Rb1 could serve as a lead compound for developing targeted inhibitors to suppress Twist1 hyperactivation and biosynthesis, thereby providing novel therapeutic strategies for ventricular remodeling HF.]
Comments 11: [In the conclusion section, the authors should reinforce the potential applications of their outcome in clinical settings contributions more explicitly.]
Response 11: Thank you for pointing out the modification. We have combed and revised the manuscript as following in (page 24, paragraph 5). [In summary, this study confirmed that ginsenoside Rb1 is a compound that inhibits ventricular remodeling in HF by improving energy metabolism disorder in HF. Ginsenoside Rb1 has the effect of maintaining myocardial mitochondrial energy metabolism in rats with HF, reducing myocardial fibrosis, and thus improving cardiac function in rats. The mechanism of ginsenoside Rb1 in treating HF is achieved by regulating the Twist1/PGC-1α/PPARα signaling pathway. This study clarifies the molecular mechanism of ginsenoside Rb1 in inhibiting HF, and suggests that Twist1 may become a clinical diagnostic marker of HF. Twist1 expression levels reflect HF-related myocardial fibrosis and therapeutic response. As both a predictive biomarker and therapeutic target, Twist1 provides critical support for monitoring treatment efficacy, and implementing precision antifibrotic interventions in HF management.]
Comments on the Quality of English Language: The English could be improved to more clearly express the research.
Response: [The manuscript underwent professional language polishing through MDPI Author Services to meet publication standards, with a focus on grammatical rigor and domain-specific terminology optimization (Service Number: english-89091).]

Round 2
Reviewer 3 Report
Comments and Suggestions for Authors
Accept in present form